# Effectiveness of Robotic Exoskeleton-Assisted Gait Training in Spinocerebellar Ataxia: A Case Report

**DOI:** 10.3390/s21144874

**Published:** 2021-07-17

**Authors:** San-Ha Kim, Jae-Young Han, Min-Keun Song, In-Sung Choi, Hyeng-Kyu Park

**Affiliations:** 1Department of Physical & Rehabilitation Medicine, Chonnam National University Hospital, Gwangju City 61469, Korea; sanha3520@naver.com (S.-H.K.); drchoiis@hanmail.net (I.-S.C.); 2Department of Physical & Rehabilitation Medicine, Regional Cardiocerebrovascular Center, Center for Aging and Geriatrics, Chonnam National University Medical School & Hospital, Gwangju City 61469, Korea; rmhanjy@hanmail.net (J.-Y.H.); drsongmk@daum.net (M.-K.S.)

**Keywords:** exoskeleton, gait training, spinocerebellar ataxia

## Abstract

Spinocerebellar ataxia (SCA) is a hereditary neurodegenerative disorder that presents as ataxia. Due to the decline in balance, patients with SCA often experience restricted mobility and a decreased quality of life. Thus, many studies have emphasized the importance of physiotherapies, including gait training, in SCA patients. However, few studies have examined the effectiveness of robotic gait training in SCA. Here, we report the therapeutic outcomes of exoskeleton-assisted gait training in a patient with SCA. A 23-year-old woman with SCA participated in a gait training program using a powered lower-limb robotic exoskeleton, ANGELLEGS. The 8-week training program consisted of standing training, weight-shifting exercises, and gait training. Several measures of general function, balance, gait, and cardiopulmonary function were applied before, after, and 4 weeks after the program. After the program, overall improvements were found on scales measuring balance and gait function, and these improvements remained at 4 weeks after the program. Cardiopulmonary function was also improved 4 weeks after the program. Robotic exoskeleton gait training can be a beneficial option for training balance, gait, and cardiopulmonary function in SCA.

## 1. Introduction

Spinocerebellar ataxia (SCA) is a collective term for neurodegenerative disorders with autosomal dominant inheritance that present with ataxic symptoms. It is a highly heterogeneous group of disorders, including more than 40 genetically distinct subtypes. These subtypes are identified by numbers assigned in chronological order as the causative genes were identified [1,2]. The reported global prevalence of SCA ranges from 0 to 5.6 per 100,000 individuals, and 2.7 per 100,000 individuals on average. The incidence of SCA differs in accordance with the type and region [1,3,4].

Most patients with SCA complain of a progressive loss of balance and coordination, accompanied by slurred speech [1]. Restricted mobility and communicative functions are common in SCA patients. SCA can impair patients’ quality of life and activities of daily living, and ultimately lead to premature death [5,6].

SCA may be present when ataxic symptoms progress without evidence of an acquired cause; genetic tests are used for confirmatory diagnosis. A known history of SCA and certain clinical symptoms are important when selecting the appropriate molecular genetic tests. Brain magnetic resonance imaging, nerve conduction studies, and cognitive function tests can also be helpful [1].

There is no consensus as yet regarding the optimal pharmacological treatments for SCA. Some reports have suggested the beneficial effects of riluzole and valproic acid, but this has not yet been confirmed [7,8]. Instead, supportive treatments are employed to maintain function [9]. Supportive treatments include physiotherapy (PT), occupational therapy, speech therapy, and addressing neurological symptoms unrelated to ataxia. Especially for PT, balance and coordination training, strengthening exercises, and gait training are recommended [10]. 

Recently, various robotic exoskeletons have been introduced to satisfy individualized purposes, and lower limb exoskeletons are one of the major classes of these exoskeletal robots. The lower limb exoskeletal robots can be broadly categorized into three types by their intended use: human performance augmentation exoskeletons for increasing physical capabilities, assistive devices for individuals with disabilities, and therapeutic exoskeletons for rehabilitation. Among them, therapeutic rehabilitative robots make it possible to train an individual’s neuromuscular system for recovery of impairment [11]. The application of rehabilitative robots reduces the burden of therapists and enables the achievement of quantitative data of training and recovery [12]. Additionally, this technology enables us to decrease the personnel burden during long-term treatments [13]. With multidisciplinary integration of sensing, control, information, computer science, bionics, and medicine, lower limb exoskeletons provide the opportunity for gait assistance and training for patients. These rehabilitative robots are predicted to have a predominant role in future rehabilitation therapy [12].

Robotic rehabilitation using exoskeletons is an emerging field, but in most studies, gait-assistive exoskeletons are mainly used for patients with stroke or spinal cord injury. To date, there are few studies applying wearable exoskeletons in SCA patients as an assistive device. However, no studies have reported the effects of rehabilitative training using an exoskeleton in SCA. Here, we report changes in the clinical symptoms and cardiopulmonary function of a patient with SCA achieved using a powered lower-limb exoskeleton.

## 2. Case Description and Methods

### 2.1. Case Description

A 23-year-old woman with SCA type 7 diagnosed 6 years ago visited our clinic. Her chief complaints were postural instability and gait disturbance. She was fully conscious and reported no other underlying conditions. She was 167 cm tall and weighed 55.3 kg. The muscle strength of both lower extremities was graded as 4 (normal = grade 5). Contracture of major joints and spasticity were not checked. On the first visit, the patient could only walk short distances using a walker due to body sway. For symptom management, a rehabilitative intervention program was established.

### 2.2. Methods

#### 2.2.1. Training Program

An 8-week intervention program was prepared for the patient. The training program involved three 30-min sessions per week for 8 weeks, for a total of 24 sessions. Each session consisted of three parts: standing training (5 min), weight-bearing and weight-shifting exercises (5 min), and overground walking in a flat hallway while wearing the exoskeleton (20 min). The walking speed during training was adjusted according to the patient’s natural pace. All training sessions were conducted in our rehabilitation center under the supervision of a physical therapist (Figure 1).

#### 2.2.2. Robotic Exoskeletal Device

The ANGELLEGS device (SG Robotics, Seoul, Korea) was used as a gait-assistive exoskeletal device. This exoskeletal device weighs approximately 10 kg and consists of hip, knee, and ankle segments and a controller box located in the individual’s back. Only the hip and knee joints are motorized and actuated. 

For individualized gait assistance in accordance with the patient’s gait phase, both force and kinematic sensors were employed. Pneumatic force sensors in both soles detected the pressure of toes and heels. As the patient started to walk, this ground force sensors sent the ground force data to the controller box and sensed the initial swing phase and initial contact phase of gait cycle. On the other hand, the kinematic sensors located in the actuation units detected the joint angle and enabled us to identify the appropriate timing for torque assistance.

Using the ground force and angular data as an input signal, the actuator can provide a predefined assistive torque tailored for the individual patient based on their gait phase and pattern. The magnitude and duration of assistive torque can be modulated depending on the individual’s functional status. Each motor can generate a continuous assistive torque of 24.3 Nm.

An additional noteworthy feature of ANGELLEGS is the low impedance of joint motor. Unlike many other overground exoskeletons with a large actuator friction, the ANGELLEGS reduces the resistive torque of actuation unit up to 0.5 Nm and allows it to prevent unexpected resistance while wearing the device. 

ANGELLEGS can be used in various individuals with the partial impairment of gait function: elderly, stroke, spinal cord injury, neuromuscular diseases, etc. Individuals with cognitive impairment, pregnancy, or other critical medical conditions need to be excluded. In our case, we applied ANGELLEGS in an SCA patient who could walk partially with physical assistance.

#### 2.2.3. Functional Assessments

Several functional assessments were performed before the program (T0), after the program (T1), and 4 weeks after the end of the program (T2). The Manual Muscle Test (MMT), Modified Ashworth Scale (MAS), Korean-Modified Barthel Index (K-MBI), Berg Balance Scale (BBS), Scale for the Assessment and Rating of Ataxia (SARA), European Quality of Life 5-dimension scale (EQ-5D), a 10-m walking test, a timed up-and-go test, and a three-dimensional gait analysis were implemented by a physical therapist and an occupational therapist. In addition, three-dimensional dynamic posturography (PRO-KIN system; TecnoBody Srl, Dalmine, BG, Italy) was used to examine static stability. The patient was asked to stand on both feet on the instrument’s tilting board for 30 s. While the patient tried to stand on the board without moving, movement of the center of pressure (COP) was recorded, and the average position of the COP was measured along the anterior–posterior (AP) and medial–lateral (ML) axes. The standard deviation (SD) of the COP was our main focus, as it indicates body sway. To assess cardiopulmonary function, we asked the patient to walk along a 10 m aisle 10 times at a comfortable pace while wearing the portable gas analyzer (model K4B2; COSMED Srl, Rome, Italy), and VO_2_ levels and metabolic equivalents (METs) were estimated.

## 3. Results

There were no major adverse events during the 8-week intervention. The changes in general assessment results, posturography, gait pattern, and cardiopulmonary function are shown in Table 1. The SARA scale score improved from 25 (T0) to 23 (T1), and a mild improvement was noted in the K-MBI (especially in transfer category) and BBS. The posturography result showed a decrease in the SD of COP in both the AP and ML axes at T1 compared to T0 (Figure 2).

The 10-m walking test time decreased from 38.12 (T0, comfortable pace) and 34.50 s (T0, fast pace) to 24.30 (T1, comfortable) and 21.88 s (T1, fast). The timed up-and-go test time also decreased, from 31.94 (T0) to 25.97 s (T1). There was no significant change in gait pattern.

The changes were largely maintained at T2. The scores for the general assessment at T2 were the same as those at T1. It is noteworthy that the patient walked faster at T2 compared to T1. The gait speed and stride length increased from 47.3 cm/s and 64.2 cm (T1) to 57.0 cm/s and 74.9 cm (T2), respectively. The 10 m walking test and timed up-and-go test times also decreased at T2. VO_2_ and MET values decreased at T2 compared to T1 at the higher gait speed.

## 4. Discussion

The loss of balance and coordination in SCA results from the damage of the central nervous system, including the cerebellum and spinal cord. Although ataxia is a representative symptom in most SCAs, the pathophysiology of each type varies widely. With its diversity and rarity, there is no well-established treatment protocol for SCA. For SCA patients, supportive care, including PT, is an essential treatment, with some medications also being potential options [9]. To achieve a similar therapeutic outcome, a robotic exoskeleton, ANGELLEGS, was used for gait training in this case. It is known that by assisting or resisting the user’s movements with a robotic exoskeleton, the impaired muscular or nervous system can be properly stimulated. As a result, enhancement of the impaired function can be predicted [11,12,14].

Winfried et al. reported the long-term effects of coordinative training in patients with degenerative cerebellar disease. In that study, 14 patients with degenerative cerebellar disease underwent 4 weeks of intensive training. The post-training results showed overall improvements on balance scales, including a decrease in the SARA score of 4.4 points (range from 2 to 7.5) on average. Although these improvements slightly declined with the disease progression, benefits still remained at 1 year compared to before training [10]. Fonteyn et al. reported a study of treadmill gait training in patients with cerebellar degeneration showing 0.4 points of decrease in the SARA scale on average [15]. Other review articles on the effects of rehabilitative interventions have also emphasized the importance of rehabilitative training in ataxia patients [16,17,18]. In our case, we adopted exoskeletal gait training as a PT program, and the results coincided with those of preceding studies of classical rehabilitation. The SARA score decreased by two points after 4 weeks of intervention, suggesting a similar achievement compared with previous studies. The balance scale score, BBS improved, and the SD of COP decreased in posturography, reflecting improved static and dynamic stability. With these improvements in balance function, we found that the patient could walk faster and with an increased stride length. The cadence, gait speed and stride length of both legs increased, and less time was taken in a 10-m walking test and a timed up-and-go test. Furthermore, these positive changes persisted for 4 weeks after the end of the program.

In terms of cardiopulmonary function, as the patient gained the ability to walk faster at T1, the oxygen demand increased compared to T0. At T2, the patient consumed less oxygen while walking at a faster gait speed compared to T1, indicating improved oxygen consumption efficiency. This improvement might be related to the increase in mobility. The improved balance resulting from 4 weeks of training may also have increased her physical activity level in daily life, resulting in enhanced cardiopulmonary function. Paul et al. reported the clinical effectiveness of unsupervised, low-budget home-based training in a randomized crossover trial. In that study, healthy middle-aged subjects who engaged in home-based exercise showed meaningful improvements in peak VO_2_ level, heart rate during resting and exercise, and stroke volume [19]. Home-based exercise was also effective in patients with stroke or heart failure [20,21,22]. Considering the improvement of K-MBI, especially in the transfer category, our patient’s cardiopulmonary function may have improved not only due to the intervention, but also due to her enhanced mobility.

Tsukahara et al. reported preceding studies about the effect of robotic exoskeleton in SCA patients as an assistive device. A wearable robotic system called “curara” was used, and it was composed of actuators at both hip and knee joints, a switch box and a controller box located on the back. With this gait assist system, patients could walk more smoothly with an increased harmonic ratio while wearing the exoskeleton compared to the gait without the exoskeleton. There were no major changes in gait speed, stride length, and cadence in most patients [23,24].

Similar to these previous studies, we adapted wearable robotic exoskeletons for the gait of SCA patients. Moreover, unlike these previous studies, we conducted a rehabilitative intervention, not only evaluating the momentary effect of the exoskeleton while wearing it, but we also determined if weeks of exoskeletal gait training could make a difference in functioning while not wearing the device. As a result, after 8 weeks of the training program, the patient showed overall improvements in functional outcomes without wearing the exoskeleton. Dynamic balance, static balance, and cardiopulmonary function were enhanced. Gait speed and stride length also increased, which differed from previous studies. In summation, beneficial effects in an SCA patient’s function are expected with and without the exoskeleton under proper rehabilitative training.

In robotic exoskeletal rehabilitation, human–robot integration plays a key role in better training outcomes [12]. The robotic exoskeleton needs to be controlled based on the human motion intention to provide appropriate force for the patient and obtain an adequate training effect. However, the method for precisely detecting human motion has not yet been established, so there have been diverse efforts to sense human motion and use it as an input signal of the exoskeleton. A mechanical sensor sensing position, force, or torque is commonly used. Biological signals can also be used by sensing the signals of electromyography or electroencephalography [11,12,25]. The ANGELLEGS in our case used two kinds of mechanical sensors for synchronization. While sensors in both soles could detect the ground contact force, the kinematic sensors in each motorized joint measured the joint angle. Further, by reducing the resistive torque during operation, proper assistive force was provided at the proper moment.

There are some limitations of this case report. First, although this study determined some potential benefits of exoskeletal gait training in an SCA patient, we cannot confirm that exoskeletal training is effective in all SCA patients with this one case. The clinical features of SCAs differ massively in each individual according to type and pathophysiology. However, considering the lack of studies, this case suggests the feasibility of adopting exoskeletal rehabilitation in SCA patients. Further research is needed to conclude the clinical practice of robotic exoskeletal gait in SCA patients. Second, as other overground exoskeletons, the assistance force during gait training using ANGELLEGS was focused on the gait motion in the sagittal plane. The gait motions in the coronal and the transverse plane also need to be considered for personalized assistance. Further studies about sensing, controlling, and proper power generation in other planes are required under the understanding of the natural gait of humans.

To our knowledge, this is the first study to report the benefits of robotic exoskeletal rehabilitative training in SCA patients. With the robotic exoskeleton, we could provide appropriate support for the patient during gait training, and as a result, overall improvements were shown in balance function and cardiopulmonary function.

## 5. Conclusions

This case suggests the positive effects of exoskeletal gait training in SCA patients. Considering the diversity of types and clinical features of SCA, larger studies of additional types of SCA would be helpful to predict the clinical efficacy of robotic exoskeleton training in SCA.

## Figures and Tables

**Figure 1 sensors-21-04874-f001:**
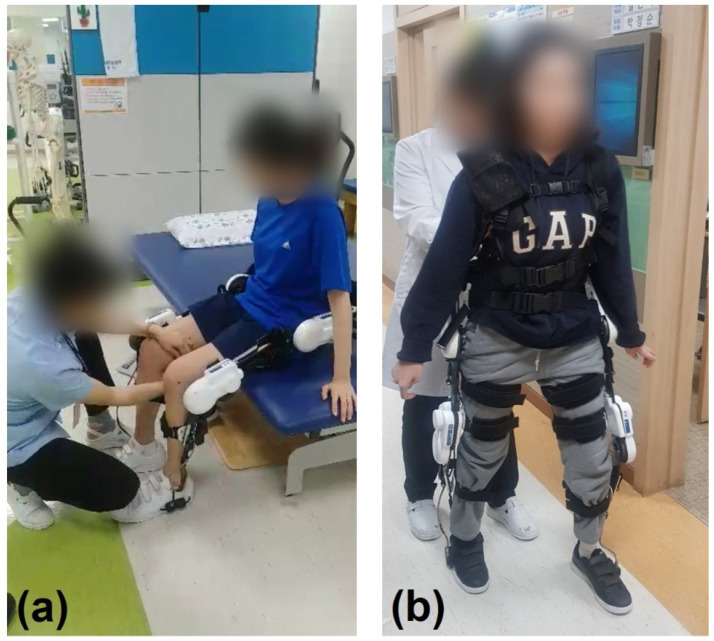
Robotic exoskeleton-assisted overground gait training. (**a**) Standing training wearing ANGELLEGS; (**b**) gait training wearing ANGELLEGS.

**Figure 2 sensors-21-04874-f002:**
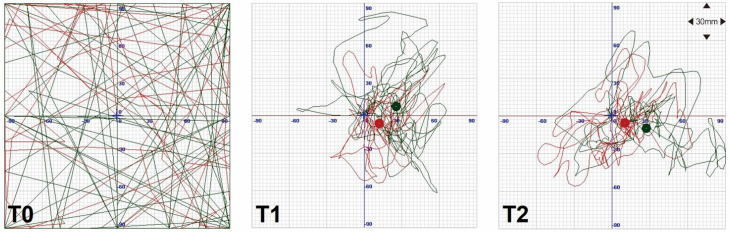
Movement of the center of pressure (COP) with the eyes open (red line) and closed (green line). Before training (**T0**), the patient could not maintain a stationary position, and the COP moved in a disorderly fashion whether the eyes were open or closed. After (**T1**) and 4 weeks after the end of the program (**T2**), the patient could maintain a relatively static COP compared to T0.

**Table 1 sensors-21-04874-t001:** Functional assessment.

	T0	T1	T2
General Assessment			
MMT (right leg/left leg)	(4/4/4)/(4/4/4)	(4/4/4)/(4/4/4)	(4/4/4)/(4/4/4)
MAS (right ankle/left ankle)	Gr 0/Gr 0	Gr 0/Gr 0	Gr 0/Gr 0
K-MBI	69	73	73
BBS	19	23	23
SARA	25	23	23
EQ-5D	8	8	8
Posturography—SD of COP AP/COP ML (mm)			
open eyes	389/404	20/14	17/17
closed eyes	446/495	30/20	20/29
10-m walking test (s)			
comfortable pace	38.12	24.30	18.81
fast pace	34.50	21.88	16.20
Timed up-and-go test (s)	31.94	25.97	20.91
Gait analysis			
cadence (steps/min)	78.8	89.8	92.2
speed (cm/s)	40.8	47.3	57.0
stride length (cm)	62.6	64.2	74.9
right/left stride length (cm)	63.2/61.9	64.5/63.9	73.7/76.1
right/left step length (cm)	27.6/35.0	24.5/39.2	32.3/42.6
step width (cm)	16.2	17.5	16.7
Oxygen consumption during comfortable walking			
VO_2_ (mL/min)	940.463	1274.272	1257.204
MET	5.229	7.13	7.04

MMT = Manual Muscle Test; MAS = Modified Ashworth Scale; K-MBI = Korean-Modified Barthel Index; BBS = Berg Balance Scale; SARA = Scale for the Assessment and Rating of Ataxia; EQ-5D = European Quality of Life 5-dimension scale; SD = standard deviation; COP = center of pressure; AP = anterior–posterior; ML = medial–lateral; MET = metabolic equivalent of task. T0, before the training program; T1, after the training program; T2, 4 weeks after the end of the training program.

## Data Availability

The data presented in this study are available on request from the corresponding author on reasonable request.

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
