# Peer review of "Effectiveness of Robotic Exoskeleton-Assisted Gait Training in Spinocerebellar Ataxia: A Case Report"

_sensors, 2021, doi:10.3390/s21144874_

Round 1
Reviewer 1 Report
The paper presents a case report regarding a robotic exoskeleton used in gait training for a patient suffering from spinocerebellar ataxia. The study reports an eight week intervention with 30 minutes sessions / week.
The case report is well structured and written however there are some suggestions that may improve the quality of the case report.
- Please add if possible, measuring units for the first part of the table 1.
- Please improve the quality of the figure 2.
- Are the authors able to provide a comparison between the robotic device for SCA and classical rehabilitation? In order to assess the success of the rehabilitation using the robotic system , at least same results as classical rehabilitation should be obtained.
- Is there enough one patient to validate the effectiveness of the robotic rehabilitation in SCA patients?
Best regards
Author Response
On behalf of all the authors, I am submitting a revised manuscript entitled " Effectiveness of Robotic Exoskeleton-assisted Gait Training in Spinocerebellar Ataxia: A Case Report". We very thank you for investing your valuable time in our article. We sincerely hope to publish our article in SENSORS.
We all authors appreciate for your detailed advices. We could revise and complement our manuscript in more detail with your thoughtful comments as following.
- Please add if possible, measuring units for the first part of the table 1.
Answer- Thank you for your kind comment. As you commented, we put the measuring units in the first line of each categories if possible.
- Please improve the quality of the figure 2.
Answer- Thank you for detailed advice. We improved the image quality of the figure 2. Also we additionally added the dimensions of X-axis and Y-axis for better understanding.
- Are the authors able to provide a comparison between the robotic device for SCA and classical rehabilitation? In order to assess the success of the rehabilitation using the robotic system, at least same results as classical rehabilitation should be obtained.
Answer- Thank you for your precise comment. There are several studies about the improvements of SCA patients after classical rehabilitation. Winfried et al. reported the long-term effect of coordinative training, showing 4.4 points (range from 2 to 7.5) decrease of SARA scale on average. A treadmill gait training in Fonteyn et al.’s study showed 0.4 points of decrease of SARA scale on average. Similarly, the patient in our case showed 2 points of decrease in SARA scale and this improvement seems close to previous studies. We supplemented these contents and reference in ‘Discussion’ part at line 179.
- Is there enough one patient to validate the effectiveness of the robotic rehabilitation in SCA patients?
Answer- We appreciate for your considerate comment. As you commented, we cannot confirm that robotic rehabilitation is effective in all SCA patients with this one case. The clinical features of SCAs differ massively in each individuals according to type and pathophysiology. However, considering the lack of studies, this case suggests the feasibility of adopting robotic exoskeletal rehabilitation in SCA patients. We supplemented this content in ‘discussion’ part as study limitation at line 237.

Reviewer 2 Report
This case reports introduces the use of ANGELEGS exoskeleton in case of a SCA patient. The findings and results of study however interesting and it can highlight the potential benefits of PT extended with robotic rehabilitation in case of the mentioned patient population, the report should be re-written before acceptance, I recommend it for major revision. Novelty is questionable, based on literature research.
My opinion in details
General impressions:
English and style should be significantly improved
There are several typos in the text.
Previous internationally recognized studies have not been critically cited.
Synergies between the main paragraphs are missing from the manuscript.
Abstract:
„Background”, „Case description, Discussion” – these titles are not necessary in the text. The type, name of exoskeleton should be mentioned in the abstract, in order to help further researches on the field (especially: meta-analysis, review writing).
Introduction:
The data regarding the prevalence of SCA is based on he a 2014 study. If possible, more recent studies should be cited, including incidence as well.
The introduction of robotic therapies is too general, the advantages, disadvantage’s and other possible use cases are not fully described. Overall, the depth of the Introduction part is not satisfactory, does not contain details. I would recommend to re-write this paragraph, including some recent international studies, for example:
1.) Shi, D., Zhang, W., Zhang, W. et al. A Review on Lower Limb Rehabilitation Exoskeleton Robots. Chin. J. Mech. Eng. 32, 74 (2019). https://doi.org/10.1186/s10033-019-0389-
2.) A. J. Young and D. P. Ferris, "State of the Art and Future Directions for Lower Limb Robotic Exoskeletons," in IEEE Transactions on Neural Systems and Rehabilitation Engineering, vol. 25, no. 2, pp. 171-182, Feb. 2017, doi: 10.1109/TNSRE.2016.2521160.
3.) R.A.R.C. Gopura, D.S.V. Bandara, Kazuo Kiguchi, G.K.I. Mann, Developments in hardware systems of active upper-limb exoskeleton robots: A review, Robotics and Autonomous Systems, Volume 75, Part B, 2016, Pages 203-220, ISSN 0921-8890, https://doi.org/10.1016/j.robot.2015.10.001
4.) Luca Toth, Adam Schiffer, Miklos Nyitrai, Attila Pentek, Roland Told, Peter Maroti, Developing an anti-spastic orthosis for daily home-use of stroke patients using smart memory alloys and 3D printing technologies, Materials & Design, Volume 195, 2020, 109029, ISSN 0264-1275, https://doi.org/10.1016/j.matdes.2020.109029..
5.) Gull, M.A.; Bai, S.; Bak, T. A Review on Design of Upper Limb Exoskeletons. Robotics 2020, 9, 16. https://doi.org/10.3390/robotics9010016
Case description and methods:
There is no anthropometric data (e.g. height, weight) regarding the patient, it would be important to mention.
Since it is a Case Report with the AGELEGS device, a more detailed description would be necessary, mainly about inclusion and exclusion criteria.
VO2 should be indicated as follows. VO2
Results:
Figure 2: Dimensions to the Y and X axis should be given, in order to better visualize the changes of COP
Discussion:
Some statements and information – despite they are generally right and correct – are not necessary to give a better understanding of the study. E.g.: „Some SCAs 129 show degeneration of Purkinje neurons in cerebellum [10, 11]. On the other hand, degeneration of brainstem or basal ganglia can also be found in some SCAs [11-13”
It would be much more important, to summarize, how robotic devices can support the rehabilitation process in general.
Many of the results are not included in the discussion, conclusion parts.
In general, authors state that this study is the first regarding the use of exoskeleton in case of of SCA patients. I would highly recommend the following two studies for further evaluation, and include the most important findings in the discussion:
1.) Tsukahara, A., Yoshida, K., Matsushima, A. et al. Effects of gait support in patients with spinocerebellar degeneration by a wearable robot based on synchronization control. J NeuroEngineering Rehabil 15, 84 (2018). https://doi.org/10.1186/s12984-018-0425-4
2.) A. Tsukahara et al., "Evaluation of walking smoothness using wearable robotic system curara® for spinocerebellar degeneration patients," 2017 International Conference on Rehabilitation Robotics (ICORR), 2017, pp. 1494-1499, doi: 10.1109/ICORR.2017.8009459.
Limititaions should be highlighted, and it should be mentioned that the results are not significant.
Author Response
On behalf of all the authors, I am submitting a revised manuscript entitled " Effectiveness of Robotic Exoskeleton-assisted Gait Training in Spinocerebellar Ataxia: A Case Report". We very thank you for investing your valuable time in our article. We sincerely hope to publish our article in SENSORS.
We all authors appreciate for your detailed advices. We could revise and complement our manuscript in more detail with your thoughtful comments as following.
General impressions:
- English and style should be significantly improved
Answer- Thank you for your thoughtful comment. We revised and corrected the contents and grammar. Also, we envelope the certificate.
- There are several typos in the text.
Answer- Thank you for your kind advice. We revised the typos in overall manuscript.
- Previous internationally recognized studies have not been critically cited.
Answer- We really appreciate for your comment. With the valuable studies you recommended in following questions, we could supplement the manuscript with much more detail. If the manuscript is still missing some important studies in some parts, please feel comfortable to comment us and we will thankfully make up for it.
- Synergies between the main paragraphs are missing from the manuscript.
Answer-Thank you for your kind comment. The authors feel the manuscript got better and became synergetic with your precious advices. We sincerely hope to publish our article in Sensors with detailed supplementation.
Abstract:
„Background”, „Case description, Discussion” – these titles are not necessary in the text. The type, name of exoskeleton should be mentioned in the abstract, in order to help further researches on the field (especially: meta-analysis, review writing).
Answer- Thank you for your thoughtful comment. We removed the subtitles (Background, Case description, Discussion) in the ‘Abstract’ part. We also mentioned the name and type of the exoskeleton. Additionally, some sentences were revised for the smoothness of the abstract.
Introduction:
- The data regarding the prevalence of SCA is based on a 2014 study. If possible, more recent studies should be cited, including incidence as well.
Answer- Thank you for your considerate comment. As you recommended, we additionally searched recent review articles about the epidemiology of SCA. These recent review articles introduced the global incidence of SCA about 2.7 per 100,000 individuals, quoting the Ruano et al.’s 2014 study. Additionally, the authors introduced the different prevalence of SCAs according to type. We supplemented this content and added reference in ‘introduction’ part at line 30.
[Reference]
Salman, M.S. Epidemiology of Cerebellar Diseases and Therapeutic Approaches. Cerebellum. 2018, 17(1), 4-11. doi: 10.1007/s12311-017-0885-2. PMID: 28940047.
Klockgether, T.; Mariotti, C.; Paulson, H.L. Spinocerebellar ataxia. Nat Rev Dis Primers. 2019, 5(1), 24. doi: 10.1038/s41572-019-0074-3. PMID: 30975995.
- The introduction of robotic therapies is too general, the advantages, disadvantage’s and other possible use cases are not fully described. Overall, the depth of the Introduction part is not satisfactory, does not contain details. I would recommend to re-write this paragraph, including some recent international studies, for example:
1.) Shi, D., Zhang, W., Zhang, W. et al. A Review on Lower Limb Rehabilitation Exoskeleton Robots. Chin. J. Mech. Eng. 32, 74 (2019). https://doi.org/10.1186/s10033-019-0389-
2.) A. J. Young and D. P. Ferris, "State of the Art and Future Directions for Lower Limb Robotic Exoskeletons," in IEEE Transactions on Neural Systems and Rehabilitation Engineering, vol. 25, no. 2, pp. 171-182, Feb. 2017, doi: 10.1109/TNSRE.2016.2521160.
3.) R.A.R.C. Gopura, D.S.V. Bandara, Kazuo Kiguchi, G.K.I. Mann, Developments in hardware systems of active upper-limb exoskeleton robots: A review, Robotics and Autonomous Systems, Volume 75, Part B, 2016, Pages 203-220, ISSN 0921-8890, https://doi.org/10.1016/j.robot.2015.10.001
4.) Luca Toth, Adam Schiffer, Miklos Nyitrai, Attila Pentek, Roland Told, Peter Maroti, Developing an anti-spastic orthosis for daily home-use of stroke patients using smart memory alloys and 3D printing technologies, Materials & Design, Volume 195, 2020, 109029, ISSN 0264-1275, https://doi.org/10.1016/j.matdes.2020.109029..
5.) Gull, M.A.; Bai, S.; Bak, T. A Review on Design of Upper Limb Exoskeletons. Robotics 2020, 9, 16. https://doi.org/10.3390/robotics9010016
Answer-Thank you for your detailed comment. We thankfully read those five articles you recommended and they were very helpful in complementing the manuscript. We added the current state of lower limb robotic exoskeleton and the benefit of it. Also, we could also supplement the ‘Introduction’ at line 50 and ‘Discussion’ at line 171, 225 with these novel recent researches. If ‘introduction’ part still does not contain detailed information enough, please don’t hesitate to comment us and we are always ready to make up for it.
Case description and methods:
- There is no anthropometric data (e.g. height, weight) regarding the patient, it would be important to mention.
Answer- Thank you for your precise comment. We supplemented the height and weight of the patient in ‘case description and methods’ part at line 76.
- Since it is a Case Report with the AGELEGS device, a more detailed description would be necessary, mainly about inclusion and exclusion criteria.
Answer- Thank you for your kind comment. We revised and supplemented the details about the exoskeleton, ANGELLEGS. We also added the target individuals and excluded individuals of ANGELLEGS, and the use of it in our case in ‘case description and methods’ part from line 93.
- VO2 should be indicated as follows. VO2
Answer- Thank you for your detailed comment. We corrected the manuscript as you indicated at line 135,
Results:
Figure 2: Dimensions to the Y and X axis should be given, in order to better visualize the changes of COP
Answer- Thank you for your kind advice. We added dimensions of X and Y axis in Fig.2. Also, we improved the image quality of the figure.
Discussion:
- Some statements and information – despite they are generally right and correct – are not necessary to give a better understanding of the study. E.g.: „Some SCAs 129 show degeneration of Purkinje neurons in cerebellum [10, 11]. On the other hand, degeneration of brainstem or basal ganglia can also be found in some SCAs [11-13]”
It would be much more important, to summarize, how robotic devices can support the rehabilitation process in general.
Answer- Thank you for your detailed comment. In accordance with your advice, we removed the sentences you mentioned. And we added the sentence about how the robotic devices can support the rehabilitation process. Additionally, we also supplemented the usual considerations of general robotic rehabilitation therapy at the latter part of the ‘discussion’ part from line 171-175, 225-236.
- Many of the results are not included in the discussion, conclusion parts.
Answer- Thank you for your considerate comment. We additionally included the results in overall ‘discussion’ part in more detail.
- In general, authors state that this study is the first regarding the use of exoskeleton in case of SCA patients. I would highly recommend the following two studies for further evaluation, and include the most important findings in the discussion:
1.) Tsukahara, A., Yoshida, K., Matsushima, A. et al. Effects of gait support in patients with spinocerebellar degeneration by a wearable robot based on synchronization control. J NeuroEngineering Rehabil 15, 84 (2018). https://doi.org/10.1186/s12984-018-0425-4
2.) A. Tsukahara et al., "Evaluation of walking smoothness using wearable robotic system curara® for spinocerebellar degeneration patients," 2017 International Conference on Rehabilitation Robotics (ICORR), 2017, pp. 1494-1499, doi: 10.1109/ICORR.2017.8009459.
Answer- We truly appreciate for your thoughtful comment. With these articles you recommended, we could complement our manuscript in much more detail. We described and added references of these two studies, and we also described the similarity and differences from ours in ‘discussion’ part at line 208-224.
- Limititaions should be highlighted, and it should be mentioned that the results are not significant.
Answer- Thank you for your detailed comment. We supplemented the limitations in ‘discussion’ part from line 237.

Round 2
Reviewer 1 Report
All the comments have been answered , I have no further comments. The case report has been considerably improved.
Best regards!
Reviewer 2 Report
Dear Authors, Respected Colleagues,
Thank you very much for your prompt answer.
The quality of the manuscript has been significantly improved, all of my adressed questions have been answered. I really appreciate that my suggestions was implemented in the report and my suggestions have been considered.
In my opinion, the corrected manuscript - after minos spell-check and editing - can be published in the present form.
Respectfully yours,